# Host Cell Response to Rotavirus Infection with Emphasis on Virus–Glycan Interactions, Cholesterol Metabolism, and Innate Immunity

**DOI:** 10.3390/v15071406

**Published:** 2023-06-21

**Authors:** Molly Raque, Sergei A. Raev, Yusheng Guo, Maryssa K. Kick, Linda J. Saif, Anastasia N. Vlasova

**Affiliations:** Center for Food Animal Health Research Program, Department of Veterinary Preventive Medicine, College of Veterinary Medicine, Department of Animal Sciences, College of Food Agricultural and Environmental Sciences, The Ohio State University, Wooster, OH 43210, USA; raque.6@osu.edu (M.R.); raev.1@osu.edu (S.A.R.); guo.1288@osu.edu (Y.G.); kick.28@osu.edu (M.K.K.); saif.2@osu.edu (L.J.S.)

**Keywords:** rotavirus A, transcriptome analysis, glycosyltransferases, fucosyltransferases, sialyltransferases, cholesterol metabolism

## Abstract

Although rotavirus A (RVA) is the primary cause of acute viral gastroenteritis in children and young animals, mechanisms of its replication and pathogenesis remain poorly understood. We previously demonstrated that the neuraminidase-mediated removal of terminal sialic acids (SAs) significantly enhanced RVA-G9P[13] replication, while inhibiting RVA-G5P[7] replication. In this study, we compared the transcriptome responses of porcine ileal enteroids (PIEs) to G5P[7] vs. G9P[13] infections, with emphasis on the genes associated with immune response, cholesterol metabolism, and host cell attachment. The analysis demonstrated that G9P[13] infection led to a robust modulation of gene expression (4093 significantly modulated genes vs. 488 genes modulated by G5P[7]) and a significant modulation of glycosyltransferase-encoding genes. The two strains differentially affected signaling pathways related to immune response, with G9P[13] mostly upregulating and G5P[7] inhibiting them. Both RVAs modulated the expression of genes encoding for cholesterol transporters. G9P[13], but not G5P[7], significantly affected the ceramide synthesis pathway known to affect both cholesterol and glycan metabolism. Thus, our results highlight the unique mechanisms regulating cellular response to infection caused by emerging/re-emerging and historical RVA strains relevant to RVA-receptor interactions, metabolic pathways, and immune signaling pathways that are critical in the design of effective control strategies.

## 1. Introduction

Rotaviruses (RVs) are a leading cause, globally, of acute viral gastroenteritis in both children and production animals [1]. The global burden of this infection is estimated to be 258 million cases of diarrhea in children under the age of five, with 235,331 deaths attributable to RVA-associated disease in 2019 [1,2,3]. This remains the third-rated pathogen associated with childhood mortality and with 24% of overall deaths in piglets [3,4,5].

As a member of the *Sedoreoviridae* family, *Rotavirus* genus, this nonenveloped virus possesses 11 segments of double-stranded RNA (dsRNA) in a triple-layer viral capsid. These 11 segments encode dsRNA for structural proteins (VP1-VP4, VP6, and VP7) and non-structural proteins (NSP and NSP1-5/6) [6]. Major antigenic properties are determined by the viral capsid proteins (VPs), while NSPs are essential for viral replication and pathogenesis [6]. Two outer capsid proteins, VP4 (the protease-cleaved protein or P protein) and VP7 (the glycoprotein or G protein), are the primary focus for epidemiological and immunological studies, as they are essential for initial attachment and penetration into cells and they independently elicit virus neutralizing antibodies [7]. RVs are classified into 11 genetically distinct (established or tentative) groups as RV A-D, F-L, which are further subdivided into distinct genotypes following the binary classification system [8]. The RV genotyping binary classification system is based on the two capsid proteins and follows GxP[x] based on the sequence diversity [8]. Group A rotaviruses (RVA) are endemic globally as the most prevalent and pathogenic among the nine RV groups, accounting for >90% of RV gastroenteritis cases [7,9,10,11,12].

RVAs have been consistently recognized as a cause of diarrhea in young piglets, and there is a pattern of re-emergence of common RVA genotypes (specifically, G1 and G9) that reach a prevalence of 61–74% across different pork production systems [13,14]. Historic studies identified porcine RVA G5P[7] as the dominant G-P combination [15], while our recent research demonstrated its reduced prevalence, with G9P[13] emerging as the most prevalent genotype combination [14,16]. Furthermore, G5P[7] replication in vivo (data not shown) and in vitro [17] is more efficient than that of G9P[13] but is not associated with increased diarrhea (in piglets) or cytopathic effects in MA-104 cell culture or porcine ileal enteroids (PIEs). PIEs represent a physiologically relevant, robust, and highly controlled system that allows the evaluation of the direct impact of RVA on the host without confounding factors, including gut microbiota, that would complicate the interpretation of results [18]. Additionally, while immune cells and adaptive immune systems are absent in enteroids, epithelial cells can produce/express numerous innate immune factors, including cytokines, chemokines, AMPS, and diverse innate immune receptors [19].

RVAs have been shown to recognize several molecules on intestinal epithelial cells, such as sialic acids (SA), histo-blood group antigens (HBGA), heat-shock cognate protein (hsc70), tight junction proteins, and integrins for attachment/entry in a genotype P-specific manner [6,20]. Interestingly, our previous study demonstrated that G5P[7] and G9P[13] differentially interacted with terminal SAs—one of the major receptors for RV entry—whereby their removal (following neuraminidase treatment) significantly downregulated the replication of G5P[7] and upregulated that of G9P[13] [17]. However, it remains unknown whether these differential initial interactions result in distinct host cell gene expression profiles.

Types I and III interferons (IFNs) were shown to play a critical role in protection against RVA infection [6,21], while the role/involvement of other innate immune factors in the context of genetically distinct RVA strains remains poorly defined. In addition to the presence of receptors for RVA entry/attachment, some components of the plasma membrane, such as cholesterol, have been demonstrated to play a critical role in RVA replication [22,23]. Moreover, there is strong evidence that cellular glycans can directly bind cholesterol and alter lipid membrane dynamics and organization [24,25]. This suggests that differential interactions with host glycans are likely to be associated with variable effects of RVAs on cholesterol/lipid metabolism. Further, similar to many viruses, cholesterol was shown to be an essential factor for RVA infection, while also being an important component in intestinal physiology and antiviral responses [26,27]. These processes are attributed to the initial interaction between RVA VP4 and co-receptors found on lipid rafts and cholesterol-rich domains promoting the cleavage of VP4 into VP5* and VP8*, endocytosis, or direct membrane penetration into the host cell, followed by the assembly of RVA [28]. Although there has been evidence supporting the essential role of cholesterol in RVA replication, the relationship between RVA replication and the biosynthesis of cholesterol is unknown [6].

Thus, the main goal of this study was to dissect the host response profiles induced by the infection of PIEs with two distinct RVA strains, with an emphasis on virus–host receptor interactions, cholesterol metabolism, and immunity.

## 2. Materials and Methods

### 2.1. PIE Maintenance

PIEs were stablished previously and maintained as described [17].

### 2.2. Rotavirus A Strains

Previously collected gnotobiotic pig small intestinal contents containing RVA G9P[13] (G9P[13] strain) and RVA G5P[7] were used in the study. Intestinal contents were diluted at a 1:10 ratio in sterile minimal essential media (MEM Gibco; Life Technologies, Grand Island, NY, USA). Contents were then centrifuged at 2095× *g* for 10 min at 4 °C and the supernatants filtered through a 0.2 µm filter.

### 2.3. Rotavirus Infection of PIEs

The rotavirus infection protocol was described by Guo et al. [17]. Briefly, 1 mL of accutase cell dissociation solution (BD Biosciences, San Jose, CA, USA) was added to each well and incubated at 37 °C for 30 min. The cell numbers were determined using Cellometer Auto T4 (Nexcelom Bioscience, Lawrence, MA, USA). The multiplicity of infection (MOI) was calculated as the amount of input virus (previously determined using a cell culture immune fluorescence assay)/total number of cells in dissociated PIEs. Prior to RVA infection, PIEs were differentiated in differentiation medium for at least 4 days. A gentle cell dissociation reagent (STEMCELL) was used to remove matrigel and allow RVA to access the apical surface of the epithelial cells. RVA inocula (5 × 10^5^ fluorescent foci units/mL) were activated prior to infection using 10 µg/mL trypsin (from porcine pancreas, Sigma–Aldrich, Cleveland, OH, USA) for 30 min at 37 °C and diluted in complete medium without growth factors (CMGF-) to achieve the desired MOI (1.0). PIEs were incubated with RVAs for 1 h at 37 °C and then washed twice with CMGF- and placed in 96-well plates in triplicate and harvested at 24 hours post-infection. Mock control PIE cells were treated identically, but inoculated with CMGF- with 10 ug/mL trypsin without RVA. Plates were kept at −80 °C until RNA was extracted from homogenized cells using a MagMAX™ Viral/Pathogen Nucleic Acid Isolation Kit (Thermo Fisher Scientific, Pittsburgh, PA, USA).

### 2.4. Transcriptome Analysis

Total RNA extracted from RVA-infected PIEs (N = 6 per virus) were sent to Psomagen to perform total RNA-seq and differentially expressed gene (DEG) and gene ontology analyses. Quantification of RNA was completed by the Ribogreen (Life Technologies, Carlsbad, CA, USA) method using Victor X2 fluorometry. The integrity of RNA was completed by Agilent RNA screentape. All samples had concentrations of >36.00 ng/µL and RNA integrity number (RIN) values above 7. Raw sequence reads were quality-checked, including the removal of adapter sequences, and then aligned to *Sus scrofa* 11.1 genome release 97 (https://useast.ensembl.org/Sus_scrofa_landrace/Info/Index, accessed on 16 September 2022) using the Rsubread aligner [29]. Gene expression counts were identified from the alignment files in BAM format using Rsubread [29] and DEG analysis was performed between virus and mock infected PIE samples. The fold change for each gene was analyzed by exactTest using edgeR with the following criteria for a significant result: |fc| ≥ 2 & raw *p* < 0.05. Data quality check: For DEG analysis, if more than one count value was 0, it was not included in the analysis. BoxPlot and density were generated to evaluate the corresponding sample’s expression distribution based on percentile (median, 50 percentile, 75 percentile, maximum and minimum), raw signal (count), Log2 transformation of CPM (counts per million reads) + 1, and TMM normalization. Finally, to evaluate reproducibility between samples, the Pearson’s coefficient of the sample’s normalized values was calculated. For the range: −1 ≤ r ≤ 1, the closer the value was to 1, the more similar the samples were. An enrichment test based on Gene Ontology (GO, http://geneontology.org/, accessed on 16 September 2022) was conducted with significant gene list using g:Profiler tool (https://biit.cs.ut.ee/gprofiler/, accessed on 16 September 2022). The web-based Ingenuity pathway analysis (IPA) tool was used to conduct analysis of DEGs to identify biological functions and molecular networks affected by both viruses using Qiagen Digital Insights and Ingenuity Systems (Qiagen, Germantown, MD, USA). 

## 3. Results

### 3.1. PIE Transcriptome Response to G9P[13] vs. G5P[7] Infection

#### 3.1.1. G9P[13] Infection Was Associated with More Profound Alterations of PIE Gene Expression

To mechanistically assess the distinct effects of G9P[13] vs. G5P[7] replication on the host (PIE) gene expression profile, we used transcriptome analysis. We demonstrated that G9P[13] replication in PIEs resulted in significant modulation of the expression of 4093 genes (Figure 1), with 1933 upregulated and 2160 downregulated. In contrast, infection with G5P[7] led to modulation of significantly fewer genes—488 with 207 upregulated and 281 downregulated (Figure 1, Appendix A). These genes were associated with diverse functions, including cellular, lipid, glycan and amino acid metabolism, intestinal homeostasis, and cancer development and resistance (Appendix A). Of interest, of the 115 genes presented in the table, G5P[7] only upregulated 46, while G9P[13] only downregulated 42.

#### 3.1.2. G9P[13] Significantly Modulated the Expression of the Genes Encoding or Regulating the Availability of RVA Attachment Factors

In order to understand the host response following the variable initial RVA-target cell interactions, we conducted a targeted analysis of DEGs associated with the biosynthesis/expression of several surface molecules known to serve as RVA attachment/entry factors (integrins, tight junctions, hsc70 protein, and glycans). Infection with both RVAs only marginally modulated the expression of gene encoding integrin αvβ3 (data not shown). However, G9P[13] infection of PIEs led to a significant upregulation of tight junction protein JAM-A—*F11R* (Figure 2), while the expression of the hsc70 protein-encoding gene was downregulated. Infection of PIEs with both RVAs modulated the expression of some glycosyltransferases. Specifically, we observed a significant modulation of the fucosyltransferase-encoding gene *ABO* (enzyme involved in HBGA synthesis [30]) after infection with G9P[13] (Figure 2). In addition, infection with both strains modulated the expression of ST3Gal5 and ST8Sia5. However, only the infection with G9P[13] was associated with a significant downregulation of the expression of genes encoding ST3Gal (ST3Gal2, ST3Gal3, and ST3Gal6) and ST6Gal (ST6Gal1) sialyltransferases. In addition, significant upregulation of expression of the genes encoding ST6GalNAc (ST6GalNAc4 and ST6GalNAc6) sialyltransferase families was observed in PIEs infected with G9P[13].

#### 3.1.3. G5P[7] and G9P[13] Infection Affected Signaling Pathways Associated with Immune Responses

A direct comparison of the most-affected pathways associated with immune response indicated that infection of PIEs with G9P[13] led to a significant activation of several cytokine signaling pathways, including interleukin IL-3, IL-6, and IL-8; chemokines (such as CXCR4); molecules associated with different functions including immune regulation [“role of double-stranded RNA (dsRNA)-dependent protein kinase (PKR) in interferon induction and antiviral response” and “N-formyl-Met-Leu-Phe (fMLP) signaling in neutrophils”] (Figure 3). Further, two signaling pathways involved in immune response regulation and inflammation, s100 family and phagosome formation, were differentially modulated by G5P[7] (downregulated) and G9P[13] (upregulated). Further analysis of DEGs associated with interferon response (Figure 4) indicated a significant downregulation of retinoic acid inducible gene I (*RIG-1*) by G5P[7] (Figure 4B). This finding was consistent with the previous findings [31] demonstrating the role of RVA interactions with RIG-I in blocking of IFN signaling. In addition, the expression of another RNA helicase-encoding gene, *MDA-5*, was upregulated after infection with G9P[13] but not with G5P[7]. These data are suggestive of the G5P[7]-mediated downregulation of the initial antiviral/IFN signaling. We also observed a prominent difference in the modulation of *CCL25* gene expression. G9P[13] infection was associated with a robust upregulation of *CCL25*, while the effect of G5P[7] infection was less prominent.

Additionally, IPA identified the top significantly affected molecules and G9P[13] infection had a staggering >200 fold-change increase of ANPEP and TMIGD1, suggestive of an increased granulocyte and cell recovery response (Appendix A). Of interest, another significantly upregulated gene that was associated with the G9P[13] infection was angiotensin-converting enzyme 2 (ACE2), encoding for an important cellular receptor that plays an essential role in inflammation regulation [32]. For comparison, the highest positive fold-change for RVA G5P[7] infection was only 17-fold for THAP8 (involved in cell apoptosis regulation). Overall, infection with G5P[7] affected fewer genes encoding for various immune factors. However, G5P[7] also decreased the expression of CD274 (also known as PDL1), NCF1, and HDAC9. These genes are involved in the inflammatory response, the regulation of programmed cell death, the formation of NADPH oxidase used by neutrophils to engulf pathogenic microorganisms, and endothelial cell apoptosis/inflammatory factor expression.

#### 3.1.4. G9P[13] Affects Cholesterol-Metabolism-Related Genes

Of interest, IPA demonstrated that the infection of PIEs with G9P[13] resulted in a significant upregulation of only one pathway associated with lipid metabolism—the ceramide pathway (z activation 2.0, −log *p*-value—1.944)—while G5P[7] infection was not associated with significant changes in cholesterol-related pathways. Ceramides are important bioactive sphingolipids involved in a variety of cellular processes such as cell growth and differentiation, inflammation, and apoptosis, including the biosynthesis of oligosaccharides and, therefore, gangliosides [33] recognized by G9P[13]. In addition, we carried out a targeted comparison of DEGs associated with intestinal cholesterol absorption and intracellular cholesterol metabolism (Table 1) [34,35,36]. Our data indicated that infection with both RVA strains significantly affected multiple (G9P[13]—6, G5P[7]—3) genes associated with cholesterol uptake, while those associated with intracellular cholesterol metabolism were less affected. Importantly, while G5P[7] infection of PIEs resulted in downregulation of the expression of the genes associated with cholesterol metabolism, G9P[13] infection led to its upregulation.

In addition, we analyzed the lipid metabolism networks associated with G9P[13] (Figure 5) and G5P[7] (Figure 6). G9P[13] infection was found to downregulate the expression of ACSL4 (acyl-CoA synthetase long chain family member 4), a gene involved in inflammation, cell death, female fertility, and cancer regulation, and FASN (fatty acid synthase regulating virus entry, host IFN response) (data not shown). The RVA impact on fertility observed in our study supports a previous study performed by Ciarlet et al., where Chinese hamster ovary cells were found to be susceptible to RV infection [37]. Importantly, G9P[13] also upregulated the expression of ACSL5 (long chain acyl-CoA synthetase) and DGAT1 (the ER-localized enzymes diacylglycerol acyltransferase), both of which are involved in lipid droplet formation [38,39,40,41]. IPA analysis generated the RVA G9P[13]-associated lipid metabolism network (with 35 genes only) and, similar to the above, demonstrated the strongest activation of TM4SF20 (ceramide signaling pathway) and B3GALT2 (involved in the biosynthesis of the carbohydrate moieties of glycolipids and glycoproteins) (Figure 6). Of interest, NAAA (the ceramide-like compound that degrades bioactive fatty acid and deactivates of palmitoylethanolamide, a lipid-derived peroxisome proliferator) and HOXA7 (involved in adipogenesis and lipid sensing, among other functions) were strongly downregulated, and IPA failed to predict interactions among the most-affected genes. Further, IPA revealed that, in contrast to G9P[13], G5P[7] infection was associated with far more complex alterations in lipid signaling. It significantly affected lipid (specifically phospholipid) metabolism, cell cycle (proliferation and apoptosis), molecular transport, and small molecule biochemistry networks. Further, while at least half of the genes in the G9P[13] network were upregulated, G5P[7] infection was mostly associated with inhibitory effects.

#### 3.1.5. Differentially Modulated Canonical Pathways

Further, to identify the functional role of DEGs using IPA, we compared the major canonical pathways that were significantly affected by the two RVA strains. The results of this analysis demonstrated that infection with G9P[13] modulated a higher number of canonical pathways, to a greater extent, compared to G5P[7] (Figure 7). While only eight canonical pathways were significantly (confidence interval 95%) affected (all downregulated) by infection with G5P[7], the inoculation of PIEs with G9P[13] resulted in a significant modulation (mostly upregulation) of 71 canonical pathways with the top 20 presented in Figure 7. In addition, surprisingly, among the most affected canonical pathways in PIEs infected with G5P[7] and G9P[13], there was only one (S100 family) common between the two RVA strains (Figure 7). However, while G5P[7] infection was associated with downregulation of this signaling pathway, G9P[13] infection led to its significant upregulation.

Analysis of the G5P[7] profile indicated that infection with this virus downregulated pathways associated with the expression of tyrosine kinases (FAK), the dominant negative inhibitor of basic helix-loop-helix proteins (ID1) [46], the growth of the intestinal mucosa (growth hormone) [47], mediators involved in the regulation of epithelial structure/function (eicosanoid) [48], molecules associated with cancer development (breast cancer regulation by stathmin 1) [49] and sperm motility. In contrast, infection with G9P[13] resulted mostly in upregulation of pathways associated with mitosis/cell proliferation and differentiation (S100 family; actin cytoskeleton; cardiac hypertrophy; ABRA; UVC-induced MAPK; regulation of cellular mechanics by calpain protease, serotonin receptor; thrombin, and ILK signaling), cancer development (the role of the tissue factor in cancer, CDX gastrointestinal cancer, and endocannabinoid cancer inhibition pathways), immune responses (CXCR4), stimulation of smooth muscle contraction in the gastrointestinal tract (the endothelin-1 pathway) [50], oxytocin release by IECs (the oxytocin pathway), and increased rates of mitochondrial metabolic activity (the senescence pathway). In addition, infection with G9P[13] was associated with the downregulation of pathways related to cell proliferation/cancer development, such as mismatch repair in eukaryotes, the role of BRCA1 in DNA damage response, kinetochore metaphase, cell cycle control of chromosomal replication, and mismatch repair in the eukaryotes pathways.

## 4. Discussion

Globally, we face a prevalence of emerging RVA strains with seemingly increased interspecies transmission potential, such as G9P[13] [51,52,53,54,55]. Numerous studies evaluating RVA pathogenesis, immune response, epidemiology, and interventions often yield inconsistent results because they are not designed to account for the drastic differences of the host response to genetically diverse RVAs. In this study, we aimed to comparatively assess the cellular signaling pathways affected by G9P[13] (emerging) and. G5P[7] (historic) RVAs infection.

In sharp contrast with previous observations suggestive of more efficient G5P[7] vs. G9P[13] replication in vivo and in vitro (unpublished data and [17,56]), our analysis demonstrated that the G9P[13] infection of PIEs led to a drastically more prominent modulation of the host cell responses compared with G5P[7]. This may indicate that G5P[7] has evolved mechanisms to improve its host-fitness and is capable of more efficient utilization of the cellular replicative machinery, while evading overt host immune response.

The significant modulation of the expression of the genes encoding sialyltransferases and fucosyltrasferases and some attachment factors following G9P[13] infection suggests that this RVA could have evolved a mechanism regulating the availability of cell attachment factors, including unmasking additional receptors that can be covered by glycoconjugates. Importantly, sialyltransferases are enzymes responsible for SA transfer to nascent carbohydrates and they can regulate the availability of RVA attachment sites. Infection with the sialidase-dependent RVA strain, G9P[13], significantly modulated (mostly downregulated) the expression of the genes encoding sialyltransferases of the four major families, ST3Gal, ST6Gal, ST8Sia, and ST6GalNAc [57], which may promote its replication. Additionally, SA and HBGAs can influence one another’s presentation and expression, subsequently altering their interactions with RVAs, as was previously hypothesized for influenza A virus [22]. Consequently, these alterations can impact intestinal homeostasis and the gut microbiota composition, predisposing the host to secondary infection or non-infectious diseases [58].

Previous studies have demonstrated that cellular glycans can directly bind cholesterol and alter lipid membrane dynamics and organization [59,60]. In addition, cholesterol has been shown to play an important role in the replication of a variety of RVs [61,62]. Our data demonstrated a robust modification of expression of genes encoding proteins involved in cholesterol transport following G9P[13] infection. In addition, we indicated that G9P[13] infection of PIEs upregulated the expression of genes of the ceramide synthesis pathway, which can potentially result in altered glycolipid biosynthesis. Earlier studies indicated the importance of ceramides in the composition of membrane glycolipids [63]. Our study elaborated on the significance of lipid metabolism in cellular receptor expression and RVA infection. Furthermore, our analysis indicated an upregulation of B3GALT2—the enzyme associated with the galactosylceramide biosynthetic process—and protein O-linked glycosylation in the Golgi apparatus in PIEs infected with G9P[13]. Thus, the strong G9P[13]-mediated upregulation of the ceramide pathway may increase the availability of gangliosides recognized by G9P[13] and other RVA strains [64]. Therefore, along with the association of *ABO* with both cholesterol [65] and glycan [66] metabolism, these data provide additional evidence of the crosslink between RVA-host glycans-cholesterol/lipid metabolism.

The differential transcriptome response of the genes related to the immune function is consistent with our previously reported observations of substantial differences in the pathogenesis and immunity associated with both strains. Accordingly, G9P[13] induced a significant upregulation of the IL-6 and IL-8 signaling pathways that have been previously shown to be involved in the pathogenesis of acute RVA-associated gastroenteritis [67]. However, the role of IL-3 (also affected by G9P[13] infection) in RVA infection remains unknown. Abundantly expressed in granulocytes, monocytes, and activated macrophages, S100 proteins are expressed in IECs under inflammatory conditions [68]. The expression of one member of this family of calcium-binding cytosolic proteins [69], S100-A6, was increased in HT-29 cells infected with SA11 G3P[2] [70]. In addition, the expression of the gene encoding another member of s100 family—S100-A8—was downregulated in T84 cells after stimulation with anti-VP7 antibodies in order to evaluate mechanisms of the development of celiac disease [71]. Interestingly, the expressions of S100A8, S100A9, and S100A12 were shown to activate the production of the major mucin of respiratory tract—MUC5AC [72]. In contrast to G9P[13], infection of PIEs with G5P[7] resulted in the downregulation of some immune-related gene expression (including the S100 encoding genes), which may be suggestive of the advanced ability of G5P[7] to evade the host immune response while still maintaining high-level replication. Taking into consideration the key role of chemokines, including CCL25, in the selective recruitment of lymphocytes to IECs [73], our data indicate that G9P[13] infection is associated with a robust upregulation of CCL25 gene expression, which may lead to the more robust immune response we observed for this virus. However, more studies are needed to evaluate the difference between the immune response to G5P[7] and G9P[13] in vivo.

Apart from the potential link to T1 diabetes, RVAs have not been demonstrated to have long-term health effects, to the best of our knowledge [74]. However, our analysis indicated that infection with both RVAs used in this study was associated with mechanisms of cancer development and cardiac diseases, reproduction, and fertility. The significant perturbations in the host glycan and cholesterol synthesis and metabolism, as well as the immune function discussed above, can explain the observed association between RVA infection and the molecular signaling related to these extraintestinal and chronic disorders. Of significance, because these RVA effects are strain/genotype-specific, comparative transcriptome analysis is essential to guide experimental science and clinical/field studies to fully evaluate RVA pathogenesis, immunity, and long-term health effects.

Taken together, this study demonstrated strain/genotype-specific modulation of the host cellular response to RVA infection, further emphasizing the unique interactions between SAs and other host factors and G9P[13] and G5P[7]. Additionally, our results suggest that long-term circulation and adaptation to swine might have increased the within-host viral fitness of G5P[7] RVA, resulting in higher replication rates and a reduced transcriptome/immune response. In contrast, G9P[13] replication was associated with a robust modulation of the host transcriptome and a significant downregulation of sialyltransferase expression. Importantly, the latter can lead to decreased sialylation and the unmasking of additional attachment factors, including gangliosides. We plan to further investigate these results, using reverse genetics and CRISPR-Cas9 gene editing to mechanistically analyze G9P[13]-specific interactions with SAs to increase its replication, and to determine how to manipulate this mechanism to reduce its growing global prevalence.

## Figures and Tables

**Figure 1 viruses-15-01406-f001:**
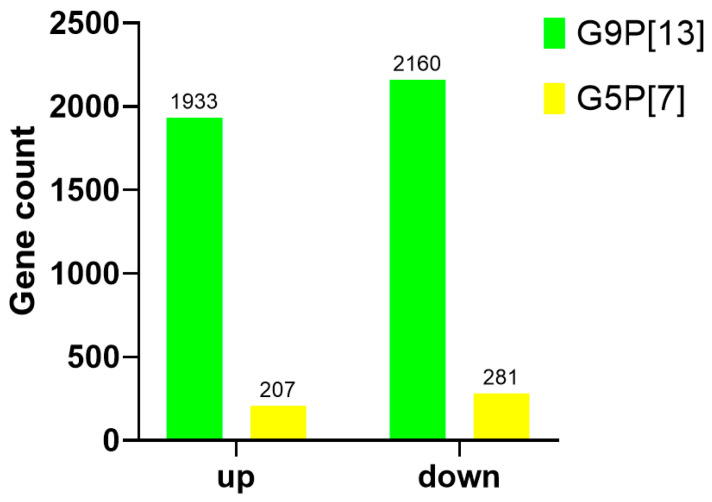
Summary of the total gene numbers that were either upregulated or downregulated (|fc| ≥ 2 & raw *p* < 0.05) based on DEG comparisons. Non-infected PIEs served as negative controls for both viruses. The total number of genes is placed at the top of the individual bars and split accordingly to represent upregulated and downregulated genes. Genes that account for these numbers can be seen in Appendix A.

**Figure 2 viruses-15-01406-f002:**
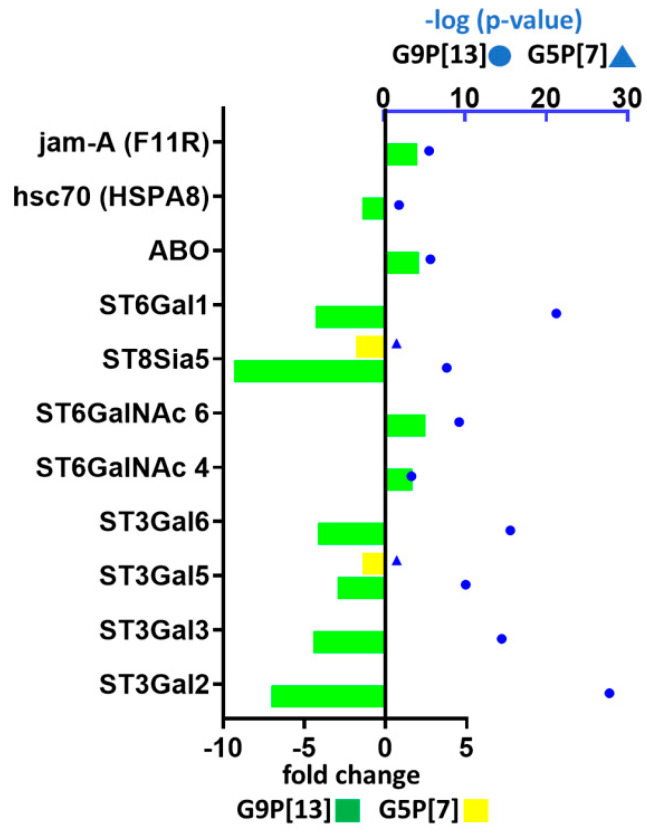
Modulation of expression of genes regulating the availability of RVA attachment factors in PIEs infected with G9P[13] and G5P[7]. The graph shows only the genes for which expression was significantly modulated (95% confidence level: *p*-value < −log1.3). Bars indicate the fold-change difference between infected PIEs and uninfected control. Blue symbols represent *p* values in log10 format.

**Figure 3 viruses-15-01406-f003:**
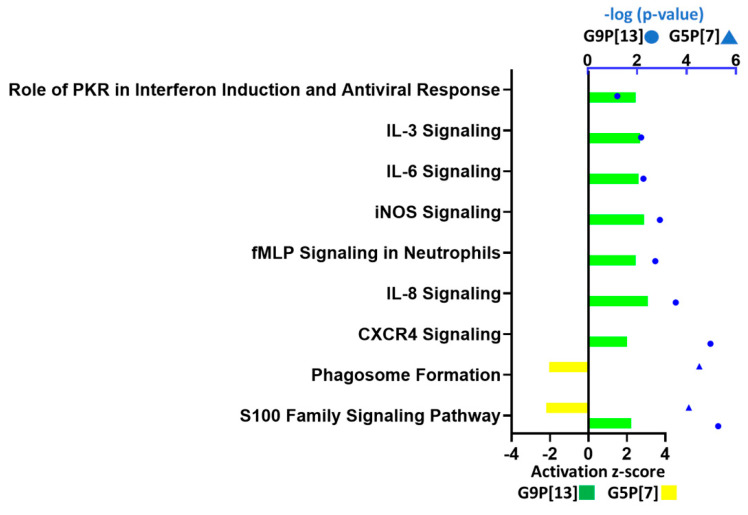
Altered molecular pathways associated with immune function in PIEs infected with G9P[13] and G5P[7]. For canonical immune signaling pathway analysis, the −log (*p*-value) > 1.3 was used as the threshold and the Z-score > 2 was defined as the threshold of significant activation, while a Z-score < −2 was defined as the threshold for significant inhibition. Bars indicate the activation z-score; blue symbols represent *p* values in log10 format.

**Figure 4 viruses-15-01406-f004:**
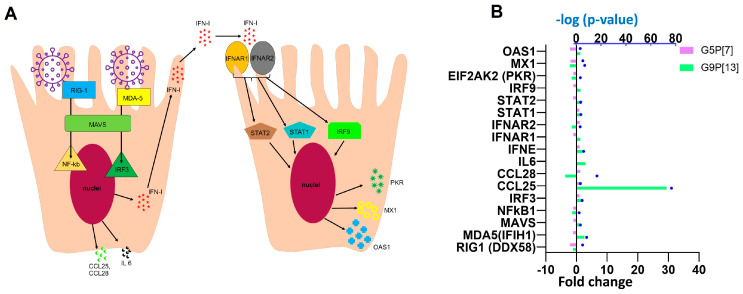
(**A**) Rotavirus interactions with the host innate immune system. RV is recognized by retinoic acid-inducible gene I (IRIG-I) and melanoma differentiation-associated gene 5 (MDA-5) receptors, followed by activation of the transcription factors IRF3 and NF-kB through signaling facilitated by mitochondrial antiviral-signaling protein (MAVS). The activated interferon regulatory factor 3 (IRF3) and nuclear factor kappa-light-chain-enhancer of activated B cells (NF-kB) modulate the expression of types I and III interferons (IFNs) stimulating the synthesis of IFN stimulatory genes. Activation of the NF-κB pathway by PRRs results in the production of proinflammatory cytokines and chemokines. IFNs are then released and bind to their respective receptors leading to activation of signal transducer and of transcription (STAT)-1, STAT-2, and IRF9 and further promoting IFN production creating an “*antiviral state*”. (**B**) DEGs associated with IFN signaling modulated in response to PIE infection with G9P[13] and G5P[7]. Bars indicate the fold-change difference between infected PIEs and uninfected control. Blue symbols represent *p* values in log10 format.

**Figure 5 viruses-15-01406-f005:**
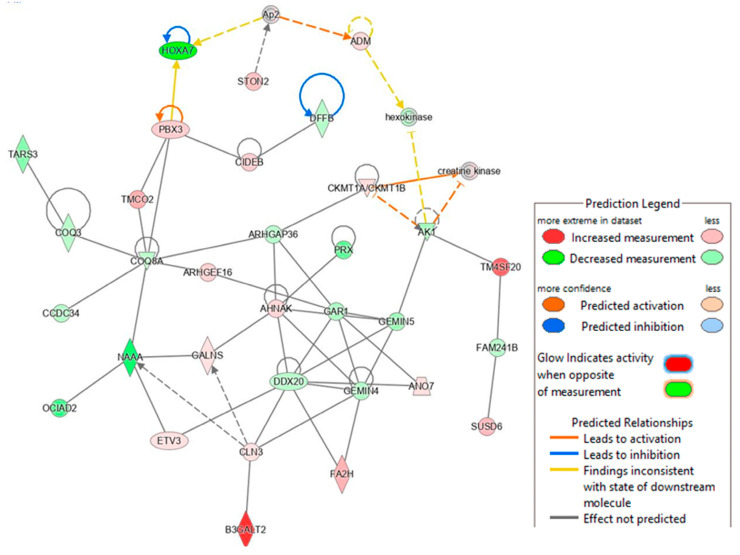
Top networks associated with lipid metabolism identified in PIEs infected with G9P[13]. IPA generates pathways using the most significant entities from the experimental data sets and connects to create a comprehensible synopsis of the analysis. In the graphical representation of a network, genes/gene products are represented as nodes, and the biological relationship between two nodes is represented as an edge (line). The intensity of the node color indicates the degree of upregulation (red) or downregulation (green), and more confidence predicted activation (blue) or inhibition (brown) of a given gene. The intensity of the edge color indicates the degree of confidence of predicted activation (orange) or inhibition (blue). The predicted pathways most affected by G9P[13] infection include the DNA fragmentation pathway of apoptosis, implemented by the increased measurement of CIDEB (cell death inducing DFFA effector B), inhibition of HOXA7, and increase of PBX3 (pre-B cell leukemia transcription factor 3) measurement. Additionally, the NF-kB (NF-KappaB) pathway is impacted here through the decreased prediction of DDX20 (DEAD-box helicase 20), suggesting a role to inhibit NF-kB activity and the activation of B cells [42]. Finally, the glycosphingolipid metabolic pathway is predicted to have high activation with impact from B3GALT2.

**Figure 6 viruses-15-01406-f006:**
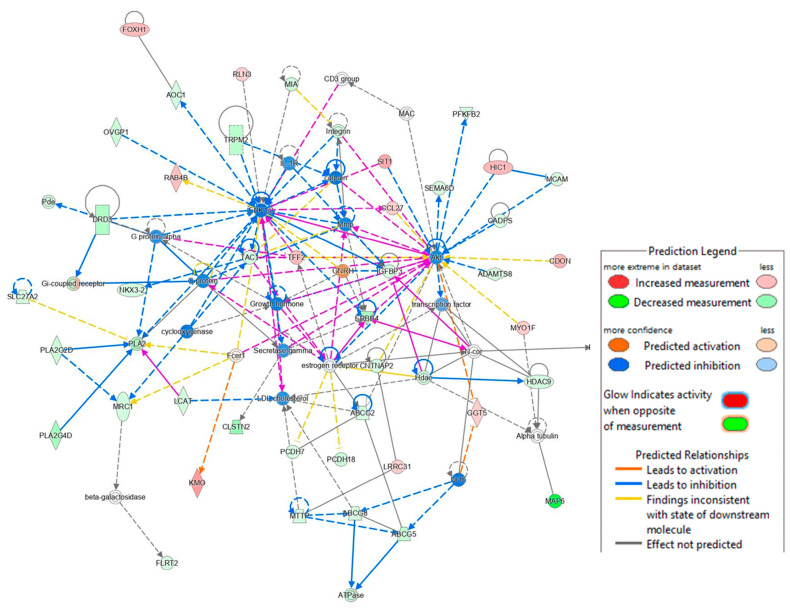
Top networks associated with lipid metabolism identified in PIEs infected with G5P[7]. IPA generates pathways using the most significant entities from the experimental data sets and connects to create a comprehensible synopsis of the analysis. In the graphical representation of a network, genes/gene products are represented as nodes, and the biological relationship between two nodes is represented as an edge (line). The intensity of the node color indicates the degree of upregulation (red) or downregulation (green), and more confidence predicted activation (blue) or inhibition (brown) of a given gene. The intensity of the edge color indicates the degree of confidence of predicted activation (orange) or inhibition (blue). G5P[7] infection was primarily associated with significant inhibition of the ERK/MAPK and phosphatidylinositol 3-kinase/Akt signaling, both of which are notable cell proliferation and survival pathways, in response to extracellular signaling. The ERK pathway serves as a “superhighway” between cellular surfaces to the nucleus of cells, where information is rapidly delivered in response to signals regarding environmental impacts and can mediate apoptosis [43]. ERK pathway degradation is associated with increased apoptosis effects [44]. Inhibition of the PI3K/Akt signaling pathway is associated with the induction of apoptosis [45].

**Figure 7 viruses-15-01406-f007:**
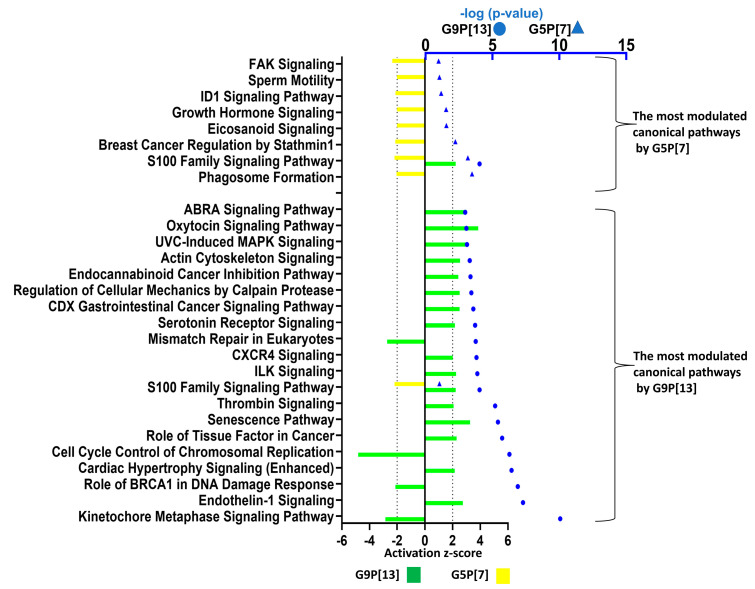
Canonical pathway analysis of PIE response following individual G9P[13] infection vs. control (non-infected PIE) and G5P[7] infection vs. control (non-infected PIE) performed using Qiagen IPA. Canonical pathways listed for G5P[7] are as follows: Focal adhesion kinase (FAK), sperm motility, ID1, growth hormone, eicosanoid, breast cancer regulation by stathmin 1, S100 family, and phagosome formation. Canonical pathways listed for G9P[13] are as follows: ABRA, Oxytocin, UVC-induced MAPK, actin cytoskeleton, endocannabinoid cancer inhibition, regulation of cellular mechanics by calpain protease, CDX gastrointestinal cancer, serotonin receptor, mismatch repair in eukaryotes, CXCR4, ILK, S100 family, thrombin signaling, senescence, role of tissue factor in cancer, cell cycle control of chromosomal replication, cardiac hypertrophy, role of BRCA1 in DNA damage response, endothelin-1, and kinetochore metaphase. Canonical pathway analysis was conducted using the IPA library and showed the most significant contributions through the input data set. For canonical pathway analysis, the −log (*p*-value) > 1.3 was taken as the threshold and the z-score > 2 was defined as the threshold for significant activation, while a z-score < −2 was defined as the threshold for significant inhibition. Bars indicate the activation z-score; blue symbols represent *p* values in log10 format.

**Table 1 viruses-15-01406-t001:** Modulation of expression of genes associated with cholesterol metabolism in PIEs infected with G5P[7] and G9P[13].

	G9P[13]		G5P[7]	
	Fold Change	−log *p* Value	Fold Change	−log *p* Value
Cholesterol uptake				
ABCG5	58.627205	89.5782	−3.995761	5.43281
NPC1L1	7.471481	38.1745	ns	ns
ABCG8	84.804386	98.0164	−2.473705	2.40029
ABO	2.057700	5.72287	ns	ns
APOE	5.547256	3.63708	ns	ns
LDLR	Ns	ns	ns	ns
MTTP	12.503798	31.0238	−2.241787	1.96767
Intracellular metabolism of cholesterol				
MOGAT2	12.087326	28.8245	−2.076138	1.57498
DGAT1	5.312413	27.28793232	ns	ns
DGAT2	Ns	ns	ns	ns
ACAT1	Ns	ns	ns	ns
ACAT2	Ns	ns	ns	ns
LPCAT3	Ns	ns	ns	ns

For DEG analysis, the −log (*p*-value) > 1.3 and |fc| > 2 were used as threshold parameters.

## Data Availability

Raw fold-change, quality check and gene ontology data are available from the authors upon request.

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
