# Peer review of "Host Cell Response to Rotavirus Infection with Emphasis on Virus–Glycan Interactions, Cholesterol Metabolism, and Innate Immunity"

_viruses, 2023, doi:10.3390/v15071406_

Round 1

Reviewer 1 Report

Host cell response to rotavirus infection with the emphasis on virus-glycan interactions, cholesterol metabolism, and innate immunity

Raque et al. compared the transcriptome responses of porcine ileal enteroids (PIEs) to two different porcine strains G5P[7] and G9P[13] infection with emphasis on the genes associated with immune response, cholesterol metabolism, and host cell attachment. I appreciate the authors' efforts to compare the transcriptome profiles for two different viruses and analyze the data in different contexts. The authors found that emerging strain G9P[13] infection led to a robust modulation of gene expression (4,093 significantly modulated genes vs. 488 genes modulated  G5P[7]). G9 P[13] mostly upregulated genes associated with diverse functions such as cellular, lipid, cholesterol metabolism, immune response, and also intestinal homeostasis compared to the historic G5 P[7] strain. The group also demonstrated that significantly modulated expression of the genes encoding RVA attachment factors occurs in the case of  G9P[13], which provides insights into novel or unexplored areas of rotavirus entry for emerging strains.

By performing a technically impressive analysis of PIE infected with two different strains -historic and emerging porcine virus, the authors suggested various pathways that are upregulated in emerging virus strains, suggesting that the historic strain evolved mechanisms to adapt to the host cell environment and developed more efficient utilization of replicative machinery for virus benefit. However, the experimental data generated from 24-hour post-infection may not be sufficient to reach that conclusion. A major drawback of this paper is the comparative transcriptome profile of two different strains based on 24-hour post-infection (single replication cycle). The data would have been stronger if the experiment is performed and compared at a later time point 72 hours (multiple replication events).

The underlying question to see the transcriptome profiles for old and emerging strains of the rotavirus is a very relevant question and data that comes from this study would be useful not only to the researchers in the rotavirus field but also to the researchers in virology and cell biology. The experiments in the paper are performed well, but many experimental details are missing, therefore the reproducibility of the study is not feasible. The conclusions of this paper are mostly well supported by data, but some aspects of the impact of this study could be further discussed. A major weakness of the paper is references- most references are outdated. Another difficulty for the readers of this paper would be understanding the experimental details. As it is a follow-up paper to Guo et al’s study (referenced as 20), many experimental details are missing, and the readers are forced to read the previous paper (Guo et al, Reference 20) to get a complete picture of the story. Therefore, providing the required details would maintain the readability of the article.

Specific comments

Introduction

1.     Most References are old and outdated. Eg; Line number 33….with 126,000 deaths in 2016….. Please include the most recent data for 2021 0r 2022. What are the most recent reports on morbidity and mortality associated with RV infections?

2.     Line 44: .......11 genetically different groups. The reference to the genetically different groups of rotavirus in line 44 should be updated according to the recent ICTV report (ICTV 2022) to ensure accuracy.

3.     Line number 54: Pls mention G5P7 used in this study is OSU strain as it is a very common porcine strain. More background information on the growth characteristic/titer of the two viruses is required to understand why both viruses behaved differently in the transcriptome analysis. From the previous publication, I understand both viruses did not replicate equally well, G9 P[13] grows better than the other. Additional info on key growth characteristics of these viruses in PIE would improve the readability and enable a better understanding of this paper.

Materials and methods

1. Line 88: It would be helpful to provide more details regarding the growth characteristics and titers of the two viruses used in the study. This information would contribute to a better understanding of why the viruses behaved differently in the transcriptome analysis. Reviewer had to read the author’s previous manuscript (reference 20) to get more fundamental details to understand the context of this paper and missing key information on viruses used in the study is a major drawback of this manuscript.

2.Clarify whether the samples collected from pigs were part of a previous study (referenced as 20) or if it is a new study. If it is a new study, please include an animal ethical report and experiment number for transparency.  

3.     Line number 98: The authors used a MOI=1 in this study. MOI=1 would not infect all the cells in 24 hours. Therefore, the transcriptome analysis in comparison to mock infection would not represent the exact expression profile in PIE due to viral replication and pathogenesis. There could be many uninfected cells in the pool, where the transcriptome profile is similar to mock cells. It would be interesting to see the transcriptome profile for a 72-hour infection and then compare that with 0-hour and 24-hour infections.

4.     Transcriptome analysis: What are the quality control steps performed to indicate the similarities and differences between the analyzed samples (N=6) and biological groups?

5.     Is there any possibility to upload the raw data to a server, which enables other researchers to access it for future comparative studies for other re-emerging strains?

 Results

The data shows the fold change of differential gene expression from transcriptome analysis, and the results section categorizes the observation under different headings such as attachment proteins, signaling pathways associated with immune response, cholesterol metabolism, and differentially modulated canonical pathways. Although it modulates multiple different genes, the data generated out of this study would serve as a great resource for many future studies such as the role of novel host proteins in RV replication. However, just transcriptome analysis may not be sufficient to convince the upregulation/downregulation profile of genes when PIEs are infected with two different strains G5P[7] and G9P[13].

During the review, it is not convincing to consider all the transcriptome results are biologically significant in the context of RV infection (For example, upregulation of cancer genes, and female fertility genes).  While the transcriptome analysis provides valuable insights into differential gene expression, it would be beneficial to complement the findings with additional assays, such as qPCR or Western blot, for the major upregulated/downregulated genes. This would strengthen the biological significance of the observed changes and provide more robust evidence

References

1.     Many references in the introduction section are outdated for a manuscript submitted in 2023.  

2.     Please consider including references relevant to this study. For example, reference 24: Role of cholesterol in Norovirus infection does not suit the statement in line number 68, which describes RVA replication.

Author Response

RE1: I appreciate the authors' efforts to compare the transcriptome profiles for two different viruses and analyze the data in different contexts. By performing a technically impressive analysis of PIE infected with two different strains -historic and emerging porcine virus, the authors suggested various pathways that are upregulated in emerging virus strains, suggesting that the historic strain evolved mechanisms to adapt to the host cell environment and developed more efficient utilization of replicative machinery for virus benefit.

AU: Thank you for the overall positive evaluation of our work and the careful assessment of the data.

RE1: However, the experimental data generated from 24-hour post-infection may not be sufficient to reach that conclusion. A major drawback of this paper is the comparative transcriptome profile of two different strains based on 24-hour post-infection (single replication cycle). The data would have been stronger if the experiment is performed and compared at a later time point 72 hours (multiple replication events).

AU: Rotavirus replication cycle is rapid and completed in about 12 hrs (Arias et al., 2004: https://www.ncbi.nlm.nih.gov/pmc/articles/PMC7126860/). Additionally, our pilot studies demonstrated that most RVA strains reach peak titers in porcine ileal enteroids by 24 hrs, while there may be a decline in viral titers by the 72 hrs post-infection. That’s why we consider 24 hours to be the optimal time for the virus harvest/transcriptome analysis. 

RE1: The underlying question to see the transcriptome profiles for old and emerging strains of the rotavirus is a very relevant question and data that comes from this study would be useful not only to the researchers in the rotavirus field but also to the researchers in virology and cell biology.

AU: Thank you for this comment.

RE1: The experiments in the paper are performed well, but many experimental details are missing, therefore the reproducibility of the study is not feasible.

AU: We concur with the reviewer and made revisions throughout to correct this (See P3, LL96-120, P4, LL136-143).

RE1: The conclusions of this paper are mostly well supported by data, but some aspects of the impact of this study could be further discussed.

AU: As above, we agree that some aspects could be discussed better and made efforts to improve the data discussion throughout (PP13-15, LL380-383; 404-406, 420, 425, 443-448, 453-460).

RE1: A major weakness of the paper is references- most references are outdated.

AU: We revised references to ensure we cite most updated/relevant ones.

RE1: Another difficulty for the readers of this paper would be understanding the experimental details. As it is a follow-up paper to Guo et al’s study (referenced as 20), many experimental details are missing, and the readers are forced to read the previous paper (Guo et al, Reference 20) to get a complete picture of the story. Therefore, providing the required details would maintain the readability of the article.

AU: We added the missing details to ensure better readability (See P3, LL96-120, P4, LL136-143).

RE1: Most References are old and outdated. Eg; Line number 33….with 126,000 deaths in 2016….. Please include the most recent data for 2021 0r 2022. What are the most recent reports on morbidity and mortality associated with RV infections?

AU: We revised references to ensure we cite most updated/relevant ones. The most recent source (https://virologyj.biomedcentral.com/articles/10.1186/s12985-022-01898-9) indicated that the global mortality due to rotavirus was estimated to be 235,331 (P1, LL33-34).

RE1: Line 44: .......11 genetically different groups. The reference to the genetically different groups of rotavirus in line 44 should be updated according to the recent ICTV report (ICTV 2022) to ensure accuracy.

AU: Updated as requested (P2, LL44-45).

RE1: Line number 54: Pls mention G5P7 used in this study is OSU strain as it is a very common porcine strain. More background information on the growth characteristic/titer of the two viruses is required to understand why both viruses behaved differently in the transcriptome analysis. From the previous publication, I understand both viruses did not replicate equally well, G9 P[13] grows better than the other. Additional info on key growth characteristics of these viruses in PIE would improve the readability and enable a better understanding of this paper.

AU: We have added the requested info (P2, LL56-59) to clarify that in fact OSU (G5P[7]) replication was most efficient despite lower binding compared with other RVA strains. This was evident that the virus titer increase from 1.5hrs (binding to host cells) to 24hrs was 2-logs and consistently significant for OSU only. G9P[13] replication (without NA treatment) was less efficient (Guo et al., 2021).

RE1: Line 88: It would be helpful to provide more details regarding the growth characteristics and titers of the two viruses used in the study. This information would contribute to a better understanding of why the viruses behaved differently in the transcriptome analysis. Reviewer had to read the author’s previous manuscript (reference 20) to get more fundamental details to understand the context of this paper and missing key information on viruses used in the study is a major drawback of this manuscript.

AU: This information has been added (P2, LL56-59 and 70-71; P3, LL114-115).

RE1: Clarify whether the samples collected from pigs were part of a previous study (referenced as 20) or if it is a new study. If it is a new study, please include an animal ethical report and experiment number for transparency.  

AU: We have clarified that enteroids were established as a part of the previous study referenced as Guo et al., 2021 and are maintained in the lab since (P3, L96). Likewise, large amounts of RVA containing intestinal contents were collected, processed and titrated previously. No additional animals were used in the current study (P3, L99).

RE1: Line number 98: The authors used a MOI=1 in this study. MOI=1 would not infect all the cells in 24 hours. Therefore, the transcriptome analysis in comparison to mock infection would not represent the exact expression profile in PIE due to viral replication and pathogenesis. There could be many uninfected cells in the pool, where the transcriptome profile is similar to mock cells. It would be interesting to see the transcriptome profile for a 72-hour infection and then compare that with 0-hour and 24-hour infections.

AU: MOI=1 was determined in our previous study as the optimum infection dose, as higher MOI would increase baseline levels (at 1.5 hrs) but not the resulting virus titers at 24 hrs post-infection. Also, it is not possible to infect all cells present in enteroids as they contain various cell types (stem, transient amplifying, Goblet, Paneth, enteroendocrine cells and enterocytes), while only terminally differentiated enterocytes are susceptible to RV infection. We appreciate the reviewer comment and plan on comparing transcriptome response kinetics in our future studies, however, it is beyond the scope of this study.

RE1: Transcriptome analysis: What are the quality control steps performed to indicate the similarities and differences between the analyzed samples (N=6) and biological groups?

AU: We have now added the following information: “Data Quality Check: For DEG analysis, if more than one Count value was 0, it was not included in the analysis. BoxPlot and Density were generated to evaluate the corresponding sample's expression distribution based on percentile (median, 50 percentile, 75 percentile, maximum and minimum) based on raw signal(Count), Log2 transformation of CPM(Counts per Million reads)+1 and TMM Normalization. Finally, to evaluate reproducibility between samples, Pearson's coefficient of sample's normalized values wascalculated. For range : -1≤ r ≤ 1, the closer the value is to 1, the more similar the samples are” (P4, LL136-143).

RE1: Is there any possibility to upload the raw data to a server, which enables other researchers to access it for future comparative studies for other re-emerging strains?

AU:  Unfortunately, the company who conducted RNA-Seq and DEG analysis for us (Psomagen) never shared FASTA/FASTQ data with us. There is a public functional genomics data repository called Gene Expression Omnibus with whom we were checking if we could deposit our data, however, they cannot accept our data without the sequence data. So, we have changed our data availability statement to “Raw fold-change, Quality check and Gene Ontology data are available from the authors upon request” (P14, L484-485).

RE1: The data shows the fold change of differential gene expression from transcriptome analysis, and the results section categorizes the observation under different headings such as attachment proteins, signaling pathways associated with immune response, cholesterol metabolism, and differentially modulated canonical pathways. Although it modulates multiple different genes, the data generated out of this study would serve as a great resource for many future studies such as the role of novel host proteins in RV replication. However, just transcriptome analysis may not be sufficient to convince the upregulation/downregulation profile of genes when PIEs are infected with two different strains G5P[7] and G9P[13].

AU: The goal of the current study was to evaluate the gene expression profiles that might explain the drastically different biological characteristics of the two strains observed in our previous studies. We would like to emphasize that there is paucity of data regarding transcriptome response to RVs and we aimed to fill this knowledge gap to inform future studies. Our plans include various functional studies to confirm the effect of the observed alterations in the expression of the host genes.

RE1: During the review, it is not convincing to consider all the transcriptome results are biologically significant in the context of RV infection (For example, upregulation of cancer genes, and female fertility genes).  While the transcriptome analysis provides valuable insights into differential gene expression, it would be beneficial to complement the findings with additional assays, such as qPCR or Western blot, for the major upregulated/downregulated genes. This would strengthen the biological significance of the observed changes and provide more robust evidence

AU: We agree with the reviewer that not all transcriptome results are biologically significant. This is not our intention to say that they are. We revised to make sure we do not overstate our findings. We conducted transcriptomics analysis because it allows to profile global and targeted gene expression and to create a comprehensive picture of cell function in response to viral infection, but most importantly we used it to answer some questions that were raised due to our previous findings and to guide our future studies and those by others. While it is not feasible to confirm the impact of most DEGs of interest in this study, and it’s beyond the scope of the study, we did analyze terminal sialic acid expression by MA104 cells following infection with G9P[13], G5P[7] and Mock and observed that G9P[13] did decrease SA expression. We decided not to include these data into the resubmission, because they are qualitative.

RE1: Many references in the introduction section are outdated for a manuscript submitted in 2023.  

AU: We revised references to ensure we cite most updated/relevant ones.

RE1: Please consider including references relevant to this study. For example, reference 24: Role of cholesterol in Norovirus infection does not suit the statement in line number 68, which describes RVA replication.

AU: We thank the reviewer for catching this error and omitted the irrelevant reference.

Reviewer 2 Report

The manuscript titled " Host cell response to rotavirus infection with the emphasis on virus-glycan interactions, cholesterol metabolism, and innate immunity" attempts to address the knowledge gap regarding the replication and pathogenesis of Rotavirus A (RVA), the primary cause of acute viral gastroenteritis in humans and animals. The study builds on the authors’ previous discovery that neuraminidase-mediated removal of terminal sialic acids (SAs) has conflicting effects on RVA-G9P[13] and RVA-G5P[7] replication.

They employed pig ileal enteroids (PIEs) as a model system to investigate the transcriptome responses to G5P[7] and G9P[13] infections, focusing on genes associated with immune response, cholesterol metabolism, and host cell attachment. The findings reveal substantial modulation of gene expression in response to G9P[13] infection, with 4,093 genes significantly affected, compared to 488 genes in the case of G5P[7]. Furthermore, the two RVA strains exhibited differential effects on signaling pathways related to immune response, with G9P[13] predominantly upregulating these pathways, while G5P[7] inhibited them. Both strains influenced the expression of genes encoding cholesterol transporters, highlighting their impact on cholesterol metabolism. Importantly, G9P[13] uniquely affected the ceramide synthesis pathway, which is known to influence both cholesterol and glycan metabolism. The study provides valuable insights into the distinctive mechanisms governing cellular responses to infection by emerging/re-emerging and historical RVA strains.

Overall, the manuscript is well-executed, and the findings add to our understanding of RVA replication and pathogenesis. The experimental design, which includes the use of PIEs as a model, improves the study's validity. The comprehensive analysis of gene expression, as well as the emphasis on immune response and cholesterol metabolism pathways, demonstrate the authors' thorough investigation. However, a few minor issues must be addressed.

1. What exactly is used for the mock infection? It is preferable to include the details in the manuscript.

2. Line 92, please double-check that the description "0.2 mm filter" is correct.

3. More discussion about whether this phenomenon and mechanism are strain or genotype specific would enhance the manuscript's impact.

Author Response

RE2: The manuscript titled " Host cell response to rotavirus infection with the emphasis on virus-glycan interactions, cholesterol metabolism, and innate immunity" attempts to address the knowledge gap regarding the replication and pathogenesis of Rotavirus A (RVA), the primary cause of acute viral gastroenteritis in humans and animals. The study builds on the authors’ previous discovery that neuraminidase-mediated removal of terminal sialic acids (SAs) has conflicting effects on RVA-G9P[13] and RVA-G5P[7] replication. They employed pig ileal enteroids (PIEs) as a model system to investigate the transcriptome responses to G5P[7] and G9P[13] infections, focusing on genes associated with immune response, cholesterol metabolism, and host cell attachment. The findings reveal substantial modulation of gene expression in response to G9P[13] infection, with 4,093 genes significantly affected, compared to 488 genes in the case of G5P[7]. Furthermore, the two RVA strains exhibited differential effects on signaling pathways related to immune response, with G9P[13] predominantly upregulating these pathways, while G5P[7] inhibited them. Both strains influenced the expression of genes encoding cholesterol transporters, highlighting their impact on cholesterol metabolism. Importantly, G9P[13] uniquely affected the ceramide synthesis pathway, which is known to influence both cholesterol and glycan metabolism. The study provides valuable insights into the distinctive mechanisms governing cellular responses to infection by emerging/re-emerging and historical RVA strains.

AU: Thank you for the overall positive evaluation of our work and the careful evaluation of the data.

RE2: Overall, the manuscript is well-executed, and the findings add to our understanding of RVA replication and pathogenesis. The experimental design, which includes the use of PIEs as a model, improves the study's validity. The comprehensive analysis of gene expression, as well as the emphasis on immune response and cholesterol metabolism pathways, demonstrate the authors' thorough investigation. However, a few minor issues must be addressed.

AU: Thank you again. We addressed the issues as detailed below.

RE2: What exactly is used for the mock infection? It is preferable to include the details in the manuscript.

AU: We have clarified in the text Mock control PIE cells were treated identically but inoculated with CMGF- with 10ug/ml trypsin without RVA (P3, LL119-120).

RE2: Line 92, please double-check that the description "0.2 mm filter" is correct.

AU: Tanks for catching our error. We have revised it to read “0.2 µm filter” (P3, L103)

RE2: More discussion about whether this phenomenon and mechanism are strain or genotype specific would enhance the manuscript's impact.

AU: We have added more discussion wherever appropriate as requested (PP13-15, LL380-383; 404-406, 420, 425, 443-448, 453-460).

Reviewer 3 Report

Raque et al investigate the transcriptome of two different rotavirus on cell explants of porcine ileal enteroids (PIE) concentrating on virus-glycan interactions, cholesterol metabolism and innate immunity.

The material and methods of RNA extraction of 6 infected PIE appears standard, and the results are clearly presented.

The (innate) immune response in the GI tract is very difficult and most likely is dependent on the microbiome present in the lumen, the epithelium likely represented to some extent by the PIE – although in an artificial environment – but dissociated from the (innate and specific) immune system. 

Type I and III interferons (IFNs) play critical roles in protection against RVA infection. However, these cytokines were not found (described) in the transcriptome. While IL-6, IL-8; chemokines and CXCR4 are likely involved in signaling but less so in initial protection like IFNs.

A plethora of other signals might be more involved in cell control and regulation.

For an interested reader in immunology, it would be desirable to get more insights - best as a figure - what this epithelial cell directly  encountering the virus represented by the porcine ileal enteroids, conveys to adjacent cells like dendritic cells, macrophages that likely produce IFN and/or activate other (immune) cells. 

Author Response

RE3: Raque et al investigate the transcriptome of two different rotavirus on cell explants of porcine ileal enteroids (PIE) concentrating on virus-glycan interactions, cholesterol metabolism and innate immunity. The material and methods of RNA extraction of 6 infected PIE appears standard, and the results are clearly presented.

AU: AU: Thank you for the overall positive evaluation of our work and the careful evaluation of the data.

RE3: The (innate) immune response in the GI tract is very difficult and most likely is dependent on the microbiome present in the lumen, the epithelium likely represented to some extent by the PIE – although in an artificial environment – but dissociated from the (innate and specific) immune system.

AU: Thank you for this comment. We have now mentioned in the text that PIEs represent a physiologically relevant, robust and highly controlled system allowing to evaluate direct impact of RVA on the host without confounding factors including gut microbiota that would complicate result interpretation. We also mentioned that while immune cells and adaptive immune system are absent in enteroids, epithelial cells can produce/express numerous innate immune factors including cytokines, chemokines, AMPS and diverse innate immune receptors (P2, LL59-64). 

RE3: Type I and III interferons (IFNs) play critical roles in protection against RVA infection. However, these cytokines were not found (described) in the transcriptome. While IL-6, IL-8; chemokines and CXCR4 are likely involved in signaling but less so in initial protection like IFNs.

AU: We have now included additional DEG analysis (new Figure 4B) that shows RVA infection effects on type I and III interferon genes. Our previous data (old Figure 3) were only showing immune signaling pathways affected that indicated that the “Role of PKR in Interferon Induction and Antiviral Response pathway” was affected

RE3: For an interested reader in immunology, it would be desirable to get more insights - best as a figure - what this epithelial cell directly  encountering the virus represented by the porcine ileal enteroids, conveys to adjacent cells like dendritic cells, macrophages that likely produce IFN and/or activate other (immune) cells. 

AU: We have included a new figure (new Figure 4A)  depicting RVA-associated immune signaling occurring in epithelial cells according to our transcriptome analysis

Round 2

Reviewer 1 Report

Thank you for submitting the revised version of the manuscript. I have thoroughly reviewed the revised version, and I appreciate the authors' prompt response to the comments and their diligent incorporation of the suggested changes in version 2.

There are a few existing typos in the draft, such as the one found in Line 96 of version 2. I believe these can be corrected during the editorial review of the manuscript.